# Gloxinia—An Open-Source Sensing Platform to Monitor the Dynamic Responses of Plants

**DOI:** 10.3390/s20113055

**Published:** 2020-05-28

**Authors:** Olivier Pieters, Tom De Swaef, Peter Lootens, Michiel Stock, Isabel Roldán-Ruiz, Francis wyffels

**Affiliations:** 1IDLab-AIRO—Ghent University—imec, Technologiepark-Zwijnaarde 126, 9052 Zwijnaarde, Belgium; francis.wyffels@ugent.be; 2Plant Sciences Unit, Flanders Research Institute for Agriculture, Fisheries and Food, Caritasstraat 39, 9090 Melle, Belgium; tom.deswaef@ilvo.vlaanderen.be (T.D.S.); peter.lootens@ilvo.vlaanderen.be (P.L.); isabel.roldan-ruiz@ilvo.vlaanderen.be (I.R.-R.); 3KERMIT, Department of Data Analysis and Mathematical Modelling, Ghent University, Coupure links 653, 9000 Ghent, Belgium; michiel.stock@ugent.be; 4Department of Plant Biotechnology and Bioinformatics, Ghent University, Ledeganckstraat 35, 9000 Gent, Belgium

**Keywords:** plant monitoring, real-time, data acquisition, sensor platform, phenotyping

## Abstract

The study of the dynamic responses of plants to short-term environmental changes is becoming increasingly important in basic plant science, phenotyping, breeding, crop management, and modelling. These short-term variations are crucial in plant adaptation to new environments and, consequently, in plant fitness and productivity. Scalable, versatile, accurate, and low-cost data-logging solutions are necessary to advance these fields and complement existing sensing platforms such as high-throughput phenotyping. However, current data logging and sensing platforms do not meet the requirements to monitor these responses. Therefore, a new modular data logging platform was designed, named Gloxinia. Different sensor boards are interconnected depending upon the needs, with the potential to scale to hundreds of sensors in a distributed sensor system. To demonstrate the architecture, two sensor boards were designed—one for single-ended measurements and one for lock-in amplifier based measurements, named Sylvatica and Planalta, respectively. To evaluate the performance of the system in small setups, a small-scale trial was conducted in a growth chamber. Expected plant dynamics were successfully captured, indicating proper operation of the system. Though a large scale trial was not performed, we expect the system to scale very well to larger setups. Additionally, the platform is open-source, enabling other users to easily build upon our work and perform application-specific optimisations.

## 1. Introduction

Plants that grow in natural or agricultural environments are exposed to substantial short-term variations in environmental conditions. For instance, the occurrence of clouds or waving leaves can modify the light environment within seconds; relative humidity and temperature can change within minutes due to precipitation. Because fitness productivity of plants often lies in their ability to swiftly respond to these highly variable conditions, studying these dynamic responses is crucial. However, research on stomatal responses and photosynthetic output often focuses on steady-state behaviour, while these conditions rarely occur in nature [1]. As a result, measurement devices are not optimised to measure this transient behaviour, while the need for monitoring the response time of plant behaviour increases in different research fields of plant science. First of all, in basic plant science where the understanding of the short-term responses to environmental variation is central [2,3,4,5]. Additionally, they are relevant in more applied phenotyping or breeding research, where the more dynamic behaviour of certain genotypes might be key to their success [1,6,7]. Furthermore, in crop management where for instance irrigation scheduling can be done using continuous measurements of stem diameter [8,9,10]. Finally, as crop and plant models become more dynamic and mechanistic, the number of parameters such as coefficients for photosynthesis and hydraulic conductance increases. In turn, this amplifies the need for dynamic data to calibrate and validate these parameter-rich models [11].

In high-throughput phenotyping, plant traits are extracted through image analysis and sometimes complemented with scoring from breeders. These techniques allow monitoring of a large number of plants both in controlled settings such as growth chambers and greenhouses, and in the field [12,13]. Depending on the set-up, the camera system can be mounted on drones [14], tractors [15], ground-based robotic systems [16,17] or conveyor belt-based systems [18,19]. These allow to extract very detailed data and specific plant traits, but their temporal resolution is generally low, ranging between once per day (e.g., a conveyor belt system) and once per week (e.g., drone flights). Consequently, they are not well suited for studying plant responses at a time scale of seconds or minutes. Other sensor types, such as porometers, cannot be permanently mounted on the plant and thus need manual intervention. Measurement devices for gas exchange (e.g., Li6800, LI-COR Biosciences, Lincoln, NE, USA) can be attached to individual plants to record short-term changes of important features like stomatal conductance, photosynthesis, and transpiration automatically, but cannot be deployed on a larger number of plants due to the cost of these devices. Other contact sensors, such as sap flow sensors or Linear Variable Displacement Transducers (LVDT), are interesting solutions for short-term continuous monitoring [8,20,21], although they require considerable time to install and might need some maintenance once installed.

Concurrent with the requirement for dynamic plant data, monitoring of a plant’s micro-environment is crucial for interpretation, considering that this is the driver for the plant’s response. Environmental data of a field trial is typically captured at a single measurement location. However, assuming the same environmental conditions for the whole field or greenhouse is not necessarily an accurate representation of reality [22,23]. Some extreme examples are the differences in microclimate between the top and bottom of a sloped field [24], shading caused by trees next to the field [25], but also differences in temperature within greenhouses, or even growth chambers [18,26].

Monitoring on the timescale of seconds or minutes, necessary to capture the dynamics of plant traits such as leaf temperature, stomatal conductance, photosynthesis, and transpiration, along with their environmental drivers, is not a trivial task [27,28,29,30,31,32]. First, interfacing and synchronising sensory readout is difficult when different sensors are combined [33]. As a result, it is necessary to rely on autonomous data acquisition systems to read out the sensors and store the data, and investigate alternative sensors that do not require manual intervention. Second, commercial data loggers such as CR1000 (Campbell Scientific, Logan, UT, USA), DL2e (Delta T Devices, Burwell, Cambridge, UK), ZL6 (METER Group, Pullman, WA, USA) or EM50 (ICT International, Armidale, NSW, Australia) are expensive, making it difficult to employ them for hundreds of sensors in large trials. Third, alternative data loggers such as single-board computers or microcontroller boards are cheap, but have limited analogue readout capabilities due to low accuracy analogue-to-digital converters (ADC) (typically in the 10 bit to 12 bit range) and do not always feature all the necessary hardware to directly interface with various analogue sensors. BeagleBone Black (BeagleBoard.org Foundation, Oakland Twp, MI, USA) is a single-board computer with analogue capabilities, and Arduino UNO (Arduino, Ivrea, Italy) is a very popular microcontroller board.

From the foregoing, we summarise that there is a need to monitor the dynamic behaviour of plant traits. To this end, a larger amount of sensors needs to be employed that is read out at a higher frequency. This has to occur cost-effectively for various purposes, including basic plant science, breeding, agronomy, and environmental monitoring. Moreover, autonomous data logging systems that do not sacrifice accuracy for cost or ease of use are key to tackle this need. We present an implementation of a data logging system that is designed to address four key needs: sensor scalability, accuracy, cost, and versatility with regards to experimental size and sensor interfaces. First, the system should easily scale to hundreds of sensors without needing a large number of hardware boards. Second, the system should provide accurate sensor readings, with limited influence of noise sources. Third, the system should be low-cost. Fourth, the system should be sufficiently versatile; it should work well in trials that monitor a handful of plants to hundreds of plants. Additionally, most common sensor interfaces should be available to connect various sensors such as light, relative humidity, temperature, and soil moisture sensors. Finally, the system should also have open-source hard- and software, enabling others to build upon this work and tailor it to a specific application.

## 2. System Architecture

To meet the key requirements introduced earlier, we selected a distributed sensing architecture. In this type of architecture, sensing is scattered across different devices, removing the need for a single measurement device that reads all deployed sensors. Nearby analogue or digital sensors are connected to the same measurement device, called node, while others are connected to other nodes. As a result, nodes only have to read out sensors in close proximity, alleviating the need for expensive low-noise cables. To facilitate the readout of data, nodes are interconnected on a linear bus. This bus is used to send measurement and configuration data. Consequently, only a single node on the bus needs to be connected to a storage device such as a computer or Raspberry Pi. The overall architecture is depicted in Figure 1.

In challenging conditions such as those present in the field, a robust and high-speed bus has to be selected for the communication between nodes. The Controller Area Network bus (CAN bus) meets these requirements. It operates at a maximum rate of 1 Mbps, while being very robust. It is commonly used in vehicles for communication between microcontrollers without the need for a master controller, thanks to a fixed arbitration scheme [34]. Nodes can easily be added; the only requirement is that the first and last nodes use a 120 Ω resistor to terminate the bus. The bus speed can also be lowered to accommodate longer bus lengths up to 5000 m at 10 kbps [35].

A Universal Synchronous Receiver-Transmitter connection (UART-connection) is used to interface between the data storage device and the node connected hereto. This computer also provides an interface to configure the nodes.

Each sensor node is comprised of a control board (named Dicio), and one or more sensor boards. These sensor boards vary depending upon the application. Communication between the control and sensor boards is done using the Inter-Integrated Circuit (I^2^C) protocol. The control node controls this bus and reads each of the sensor boards at predefined time steps [36]. Optionally, a phase-locked signal can be used to achieve cycle synchronisation between sensor boards on a single control node. By default, the I^2^C protocol operates at 400 kHz, but can also be lowered to 100 kHz for sensors that do not support fast I^2^C operation.

Two I^2^C buses are available, one for the readout of the custom sensor boards through a dedicated interface connector, and one for other sensors. Sensors such as digital relative humidity and temperature sensors are connected using screw terminals.

The dual bus system accommodates varying needs. First, the CAN bus interconnects different sensor nodes, thus simplifying the readout system and data storage interconnect. Only a single node needs to be connected to the computer. Furthermore, the CAN bus is robust in noisy environments and well-suited for communication over longer distances (modifiable in the software). The maximum distance is linked to the signalling rate and can be increased at the expense of lower throughput. The system uses a single cable to deliver both signal, power, and synchronisation signals. A synchronisation signal can be used to synchronise sampling between different nodes. Second, the I^2^C protocol was selected since it is widely used for digital sensors and allows the sensors to be read directly from an Arduino or Raspberry Pi, thus facilitating faster prototyping and stand-alone usage of the sensor boards. Furthermore, most microcontrollers provide a hardware I^2^C-interface, which reduces cost and complexity. Since the interconnect distance between the sensor boards and the control board is small, there will be less noise and interference. Consequently, the requirements of this bus are a lot more tolerant.

Two boards that cover a wide range of analogue sensors have been designed: Planalta and Sylvatica; Section 3 provides more details on the available functionality. These sensor boards cover the most common analogue interfaces found in plant monitoring. Digital sensors should interface directly with the Dicio board. Dicio supports common digital interfaces, including I^2^C, Serial Peripheral Interface (SPI) and RS-232.

Several Planalta or Sylvatica boards can be connected to the same Dicio control node to increase the number of sensors that can be read while keeping the amount of redundant hardware to a minimum. The two sensor boards, Planalta and Sylvatica, each serve a different purpose. The board named Planalta is designed for sensors that require a variable input voltage such as soil capacitance, LVDT, and impedance sensors. The measurement principle of this board relies on a digital lock-in amplifier (LIA) (see also Section 3.2). The other sensor, named Sylvatica, is designed for sensors that do not require an input signal, like most analogue temperature and relative humidity sensors, for example.

Both sensor boards have generic analogue interfaces and can easily be used with a wide range of sensors that require low input voltages and currents. Sensors should have an operating voltage between 0 V to 12 V in the case of Sylvatica and 0 V to 3.3 V for Planalta. High-power sensors such as some types of sap-flow sensors require an external power supply. An attenuator is required for both boards if the readout value can be higher than 3.3 V. To keep the hardware low-cost, only single-ended signals are supported. The Planalta board features an option to use a mid-rail referenced signal.

## 3. Measurement System Design

Some of the identified design criteria are conflicting, such as low unit cost, and accuracy and versatility. Consequently, a trade-off was made. The main cost in this system is comprised of the necessary components. As a result, the measurement system has been designed to minimise the number of analogue components, thus lowering the overall cost at the expense of more digital processing. Since the software development cost is a one-time investment, most of the signal processing and filtering are done digitally to reduces the hardware cost per unit. Furthermore, the software is easily modifiable due to a simple and generic design. This is ideal for application-specific optimisations. We selected the dsPIC33EP512MC806 (Microchip Technology Inc., Chandler, AZ, USA) as microcontroller. This is a high-performant 16-bit devices that features special signal processing instructions, direct memory access, and a CAN-interface.

The component cost is lowest if the internal ADC of the microcontroller is used. However, this is undesirable since the effective number of bits is only 11.3-bit. The objective is to design a high-accuracy system. Consequently, an external ADC is required with a generic filtering stage in front for maximum flexibility. The ADC-choice determines the analogue front-end since certain specifications have to be met to achieve maximum performance. Moreover, integrating all sensing functionality on a single board was deemed too complex and would increase the cost per board, while some of the hardware would remain unused. Therefore, two sensor boards are designed; Planalta and Sylvatica.

The Planalta board is optimised for sensors that measure modulated signals. A lock-in amplifier is well-suited for this purpose. A digital lock-in amplifier modulates a voltage (or current) based on a reference clock at a specific frequency. This signal is then deformed by the sensor-response, resulting in a new signal that has a different phase and amplitude than the original signal. An ADC whose sample points are synchronised to the same reference clock then digitises the analogue signal. The amplitude and phase can then be determined with high precision, based on the reference signal. Digital LIAs have significant advantages over their analogue counterparts, including better noise performance, phase stability and orthogonality due to the lack of temperature and frequency-dependent drift [37]. Sensors that can take advantage of this measurement principle include LVDT, impedance, and laser-based sensors.

Sylvatica is a board designed for single end-measurements. Sylvatica does not feature an analogue sine wave generator, nor a connection from the input voltage of the sensor to the ADC. Instead, it supports double the number of sensors compared to Planalta (eight sensors vs. four sensors) because more ADC channels are available for sensor outputs.

In what follows, the analogue front-end is discussed first, followed by the digital signal processing of the sampled signals.

### 3.1. Analogue Front-End Design

The front-end design focuses on simplicity and flexibility by using a generic design that is optimised digitally for the application’s needs. The driver uses well-chosen components to limit both non-linearity and noise of the circuit before digitisation.

A general overview of the analogue front-end is depicted in Figure 2. Dashed components are specific to the Planalta board. The left part of the Planalta-only circuit generates a sine wave from a pulse-width modulation (PWM) signal. By varying the duty cycle at a high frequency, a sine wave is formed after low-pass filtering. This signal is then buffered before it passes through the sensor. Buffering ensures that there is no voltage drop due to variations of the load impedance. The buffer can provide up to ±5 mA of current to the sensor. For high-power sensors, an external amplifier has to be used. On the Planalta board, the output voltage signal is also provided as an input to an ADC channel after low-pass filtering.

The circuit that connects the sensor output to the ADC has the same topology on both Planalta and Sylvatica. The output of the sensor is amplified using the PGA113 (Texas Instruments, Dallas, TX, USA) programmable gain amplifier (PGA), providing a gain between 1 to 200. This PGA can amplify with respect to an offset, on Sylvatica this offset is connected to ground potential, while on Planalta this offset can be soldered to either ground potential or 1.65 V. The signal is then low-pass filtered using a resistor-capacitor filter before digitisation. Digitising the signal as early as possible limits the amount of noise that can enter the system and increases flexibility, since the digital filtering is easy to modify.

The ADC used in this design is ADS8332 (Texas Instruments, Dallas, TX, USA), and provides a good trade-off between speed, accuracy, and cost. This ADC has a successive approximation register (SAR) architecture, produces 16-bit output data, and has an integrated multiplexer that can rapidly switch between channels, enabling up to 8 analogue signals to be sampled between 0 V to 3.3 V. The 16-bit words are well-suited for further processing by the 16-bit microcontroller. The effective ADC resolution at frequencies below 1 kHz is 14.9-bit. Additional details on the analogue front-end are provided in Appendix A.

### 3.2. Digital Signal Processing

After digitisation, the microcontroller further processes the samples to remove unwanted noise and interfering signals such as those originating from the 50 or 60 Hz power grid. The data rate coming from the ADC depends on the board and software configuration. The data rates that are possible in the current design are listed in Table 1.

The maximum sampling frequency of the ADC is 500 ksps. Though, in practice, the upper sampling speed cannot exceed 250 ksps due to limitations of the SPI module of the microcontroller. On the Planalta board, the number of active channels determines the maximum signal frequency for the LIA. The total sample rate (sum of sample rates of every channel) should never exceed 250 ksps.

The sampling frequency is always four times the signal frequency. This simplifies the mixing operation significantly, since the in-phase (I) and quadrature (Q) components of the sine and cosine are simply 0, 1, 0 and −1, and 1, 0, −1 and 0 respectively for 0 rad, π/2 rad, π rad and 3π/2 rad. These operations involve single-cycle copy and invert instructions. An overview of the whole digital processing cascade is depicted in Figure 3. The whole filter structure is replicated eight times for the four sensors and four reference signals. The decimation factor and filter coefficients of the last stage depends upon the configuration. To obtain an output frequency of 1 Hz, the decimation factor of the last stage has to vary between 2 and 20.

For the Sylvatica board, the signal frequencies of interest are typically much smaller. Biological and environmental sensor responses vary in the range of a few Hertz and below [22,27,28,29,30,31,32]. Consequently, there is no need to maximise the sampling frequency other than to limit the amount of noise. Some plant processes such as the absorption of photons by chlorophyll molecules and chlorophyll fluorescence after photon incidence occur much faster, within 1 × 10${}^{-15}$ s and 1 × 10${}^{-9}$ s respectively [38]. However, these events cannot be observed by simple sensors in a greenhouse or on the field, so they are not considered here.

To avoid active filters, the sampling frequency is increased to 10 kHz per channel. However, this shifts complexity from the analogue to the digital domain. This is desirable since it is easier to modify and optimise the software for a particular application. An overview of the digital processing is shown in Figure 4.

### 3.3. Digital Interfacing with Sensor Boards

As mentioned, I^2^C is used to read the sensor data. On this bus, one device has control and can read and write to other devices. In this setup, Dicio is the control device that reads the sensor data from the sensor boards Planalta and Sylvatica. Each device, except for the controller, has a 7-bit address that must be unique on the bus.

To ensure that only valid data are read, a ping-pong buffer system is used at the output. One buffer, buffer A, is written by the software with new data, while the other, buffer B, can be read by the user. When buffer A is full, the roles are revised. Buffer B is written, and buffer A is read. Each buffer stores one sample (this can be increased in the software) and thus has to be read every second. Note that for Sylvatica a sample consists of a single 16-bit value, but for Planalta a single sample consists of up to two or four 16-bit values: two values representing the I- and Q-components of the sensor signal and optionally the driver I- and Q-components.

## 4. Results

To validate the system, a small prototype was constructed consisting of one Dicio board, two Sylvatica boards, and one Planalta board. An experiment was conducted during ten days in a growth chamber of 1.45 × 0.77 × 1.45 m (height × depth × width) at Flanders Research Institute for Agriculture, Fisheries and Food (ILVO), Melle, Belgium with a custom-built frame of 1.00 × 0.70 × 1.10 m (height × depth × width). All lighting was mounted on this frame, including 32 LED lamps (MAS LED spot VLE D 4.9-50W GU10 927 60D, Koninklijke Philips N.V., The Netherlands) and eleven halogen lights (DECOSTAR 51 PRO 50 W 12 V 36° GU5.3, OSRAM GmbH, Germany). The experiment applied a simulated day-night cycle on two strawberry plants (*Fragaria*× *ananassa*, labelled S1 and S2) and one plum tomato plant (*Solanum lycopersicum* L., labelled T). The strawberry plants were mature plants, grown in a greenhouse at ILVO during the previous year. The main difference between them was their size, S1 had significantly more leaves and S2. The leaves of S2 were also less green than those of S1. Their pot sizes were the same. The tomato plant was a five-week-old seedling.

The two sensor boards, Sylvatica and Planalta were used to perform the readout of several contact sensors that were connected to the plants at a rate of 1 Hz. We employed four leaf thickness sensors, two leaf length sensors, one soil moisture sensor, one relative humidity and temperature sensor, and one light intensity sensor. The environmental sensors were mounted on a separate board, which was glued to a 3D-printed radiation shield. An overview of the different connections to the sensor boards and the monitored plants is depicted in Table 2. More detailed information about the different sensors is provided in Table 3.

At the start of the experiment, the plants were watered and left to stabilise for one day in the growth chamber before monitoring started. During the monitoring experiment, the plants were watered twice after visual observation of wilting of S1 just before noon on 24 December and 30 December. These time points are indicated by a dashed green line on Figure 5. A picture of the experimental setting just after the second watering time point (30 December 2019 at 10:45) is depicted in Figure 6. S1 wilted, while the other plants did not show any visual sign of wilting.

Both the leaf thickness and leaf length sensors have a strong temperature dependence, which is eliminated using a simple calibration procedure. During this calibration, the temperature was gradually increased from 10 ${}^{{}^{\circ}}$C to 32 ${}^{{}^{\circ}}$C. For the leaf length sensors, the AgriHouse Calibration Card (AH-300C) was used to calibrate. The leaf length sensors were calibrated using four reference distances.

To calculate the soil water content from the capacitance reading, the following calibration procedure was followed: first, the soil was saturated with water for five days; second, it was left to drip to remove excess moisture for 1 h; third, the soil was left to dry at ambient temperature conditions for 14 days during which the weight of the pot and sensor readouts were recorded; finally, these were combined with the dry weight and volume of the port to calculate the amount of water in the soil per volumetric unit.

The LVDT sensors, used for leaf elongation measurement, do not provide absolute values of leaf length. Therefore, the first measurement was taken to be the reference distance and was set to zero for both sensors.

A more detailed zoom of a 12 h period is of the grey area is depicted in Figure 7. The captured data contains less noise than expected from Figure 5. Moreover, the oscillatory behaviour of the sensors appeared due to the functioning of the growth chamber. It causes the environmental conditions to oscillate around a predefined setpoint, which in turn are main drivers for the plant response.

## 5. Discussion

### 5.1. Evaluation of the Experiment and Future Improvements

Figure 5 depicts the most interesting sensor data throughout the entire experiment. The visual observations of wilting are supported by the readings from the leaf thickness sensors. S1 was wilting (see Figure 6). The leaf thickness, presented in Figure 5c (blue), gradually decreased from the previous watering event on 24 December towards this time point of visual observation on 30 December at 10:45. Leaf thickness decreased from around 115 µm to 75 µm in the first drying period with decreasing soil water content down to 200 g/L. In the second drying period, soil water content decreased down to 125 g/L, resulting in a minimum leaf thickness of 53 µm. After re-watering, the leaf thickness quickly recovered. S2 was not wilting, which is supported by a more constant pattern of the leaf thickness towards this time point of observation (Figure 5c, in green). Once wilting started, there was also a clear decrease in leaf thickness for S1 to approximately 75 µm the first time and 53 µm the second time. The leaf thickness quickly recovered after watering, clearly highlighting the need for monitoring systems with high temporal resolution. Without them, it would not be possible to measure the recovery time of the leaf thickness of S1. Furthermore, there was no clear difference before and after the watering time point for S2, indicating that this plant was not perceiving drought stress in this period. S2 was probably not wilting due to its lower leaf area compared to S1, while the pot sizes were the same. As a result, we presume that the water content in the pot of S2 was still sufficient since the water content in both pots was the same when the experiment started.

For S1, the leaf elongation did not show drastic variations in response to the drying conditions, other than a small gradual growth during the experiment. The leaf elongation of the tomato plant T did not show a decrease in response to limited water availability. However, a marked increase in leaf elongation coincided with the two re-watering events, indicating that leaf elongation was slowed down before re-watering.

Besides the effects of drying and re-watering, the leaf thickness and leaf elongation sensors also demonstrated a pattern of shrinkage during the day and increased during the night. Indeed, leaf length and thickness decrease when water loss due to transpiration is not fully compensated by the water uptake, and increase when transpiration decreases and tissues are replenished with water. The pattern of swelling and shrinking was most explicit in the leaf elongation of the tomato leaf T. Additionally, the leaf thickness in plants S1 and S2 did not increase throughout the experiment, as these leaves had already reached their final leaf thickness. The elongation sensors on plants S1 and T, showed a gradual increase in leaf length, indicating that these leaves were still expanding. The elongation of the young tomato leaf was much faster than that of the leaf of the strawberry plant S1 [39,40].

From Figure 8, we can conclude that the leaf thickness is strongly influenced by the relative humidity, where the drop in relative humidity corresponds to a similar decrease of the leaf thickness around 18:35. The time offset is probably due to the heterogeneously of the air in the growth chamber. A similar effect is observed between 18:55 and 18:58. When the relative humidity decreases, water will evaporate more quickly, resulting in reduced leaf thickness. Since there is a variation of less than 1 ${}^{{}^{\circ}}$C, there is a limited effect of the temperature. These physiological responses are only detectable when a high temporal resolution is used, illustrating the need for systems such as Gloxinia. The leaf length variation remains limited. Compared to leaf thickness, leaf length features less variation on a short timescale for strawberry plants in this experiment.

As expected, soil water content consistently decreased after watering. However, slight increases at the start of the day are due to the temperature dependence of the sensor. A possible explanation is that the soil temperature was not measured, only air temperature. As a result, there is some over-compensation when the temperature changes drastically due to the slower temperature increase of the soil.

### 5.2. Design Validation and Comparison to Existing Platforms

In the introduction, four key design criteria were identified. A comparison between our design, a commercial data logger, a single board computer with analogue capabilities (BeagleBone Black), and a microcontroller platform (Arduino) is depicted in Figure 9 on a 1 to 5 scale. The higher the scale, the better the performance for this criterion.

We evaluate the requirements for two experiments: a smaller experiment where one plant is monitored closely with 15 analogue sensors, and a larger experiment where ten plants are monitored in a greenhouse with 15 analogue sensors connected to each plant. Additionally, the environment will be characterised at each plant in both experiments for temperature, relative humidity, and light intensity. In addition to analogue sensors, a digital sensor is employed that uses I^2^C to measure the temperature, relative humidity, and light intensity. To simplify the comparison, it is restricted to sensors that do not require an input waveform. All sensors are sampled every 10 s.

To evaluate the platforms for each of the criteria, we define the four measures, one for each of the design criteria. First, to assess the sensor scalability, we compare the average number of boards per sensor for each of the trials. Second, to assess the accuracy, we compare the number of bits of the output sample. While this is not the actual accuracy, this estimate provides a first indication thereof. Third, the cost is defined as the average cost per sensor of the entire readout system per platform. Fourth, versatility is assessed by qualitative comparison of the difference between the first three design criteria for the two experimental set-ups aforementioned.

The official Arduino Uno board can interface with up to six sensors on a single board, with a resolution of 10-bit. A single board costs 18.81 € (Mouser, USA, 30 January 2020). Different boards are connected using I^2^C since a robust protocol is not supported without additional hardware. As a result, three boards are needed for the first experiment and 25 for the second [41].

The BeagleBone Black Rev. C board can also interface with up to six analogue sensors per board. The resolution is 12-bit per sensors and costs 62.75 € per board (Mouser, USA, 30 January 2020). Different nodes are connected using the CAN bus. As a result, three boards are needed for the first experiment and 25 for the second [42].

Commercial data loggers are very popular and widely used by researchers in plant and environmental monitoring trials. The cost of such commercial systems is approximately 1000 to 1500 €. A single data logger can typically measure 16 sensors sequentially. And have a resolution of 12-bit to 13-bit. Thus, one data logger is needed in the first experiment and ten in the second. However, in a real set-up researchers will typically use a multiplexer to readout all sensors and keep the overall cost of the setup more manageable.

The Gloxinia system discussed here requires the Dicio control board and the Sylvatica sensor board to interface with these sensors, one Sylvatica board can measure up to eight analogue sensors and has a resolution of 16-bit. Different Dicio boards are connected using the CAN bus. A cost overview is depicted in Table 4, where a categorical separation is made. For the first experiment, one Dicio board is needed, while for the second ten boards are required. Per Dicio board, two Sylvatica sensor boards are necessary in both cases.

From the foregoing, it is clear that in terms of scalability, the Arduino performs worst due to the lack of a robust communication interface for larger distances. Commercial systems usually require expansion units, though the cable length between expansion units is usually limited. Often, this is compensated by use of expensive measurement cables with low attenuation. The BeagleBone Black and Gloxinia both support the CAN bus. Some additional hardware is required for the BeagleBone Black to interface with other CAN-enabled devices though.

The versatility of the platform depends upon the needs of the application. Therefore we compared the available sensor interfaces and the change in the number of boards needed between both experiments. The Arduino has the lowest number of interfaces, followed by the BeagleBone Black. Both lack the ability to interface with LIA-based sensors for instance. The number of boards scales similarly for Arduino and BeagleBone Black. The commercial system is the most versatile, since it has the widest range of sensor interfaces and expansion units can be added. Gloxinia’s performance is intermediate between the BeagleBone Black and commercial data logger thanks to its wider range of interfaces.

A summary of these observations is depicted in Table 5 and Figure 9. The Gloxinia platform is not the most effective on all criteria but provides the best trade-off to achieve a good score on all criteria.

### 5.3. Future Improvements and Possibilities

While not tested in the experimental setup, the system should easily scale to large trials that need to monitor sensors over large distances thanks to the CAN bus. Theoretically, there is an upper limit of approximately 400 Dicio boards that are connected to a single CAN bus, based on the differential input resistance and drive capability of the MCP2542FD CAN transmitter. However, we advise that no more than 100 Dicio nodes are connected to the same bus for error-free operation. Each Dicio node supports up to four Planalta and five Sylvatica sensor nodes. A second CAN bus has to be used in case more sensors have to be measured.

To stimulate usage of this data logging tool, both hardware and software have been open-sourced in a GitHub repository. All relevant files can be downloaded from GitHub (https://github.com/opieters/gloxinia).

## 6. Conclusions

The Gloxinia sensor platform aims to advance monitoring in fundamental and applied plant research from modelling to irrigation and crop management. Four key needs were identified: sensor scalability, accuracy, cost and versatility. The whole platform has been designed to address these needs with an open-source design. The platform comprises of individual sensor nodes that communicate with each other. Each node has a control board Dicio to which sensor nodes are connected. Sylvatica and Planalta are two sensor boards that provide an interface that matches most analogue sensors used in plant research. Digital sensors can also be connected to the control boards. Most of the application-specific optimisations are done in software, making it easier for the user to optimise for a specific application. To validate the accuracy of the system, an experimental trial was set up in a growth chamber. Environmental conditions, leaf length, and leaf elongation were successfully measured at high resolution on one tomato and two strawberry plants to validate the functionality of the system. The overall system scales well due to the multiplexed sampling of up to eight channels on Sylvatica and four on Planalta, accurate 16-bit data acquisition, lost unit cost, and distributed architecture. Consequently, the system strikes a good trade-off between these various requirements, making it well-suited for research, breeding, and precision crop phenotyping.

## Figures and Tables

**Figure 1 sensors-20-03055-f001:**
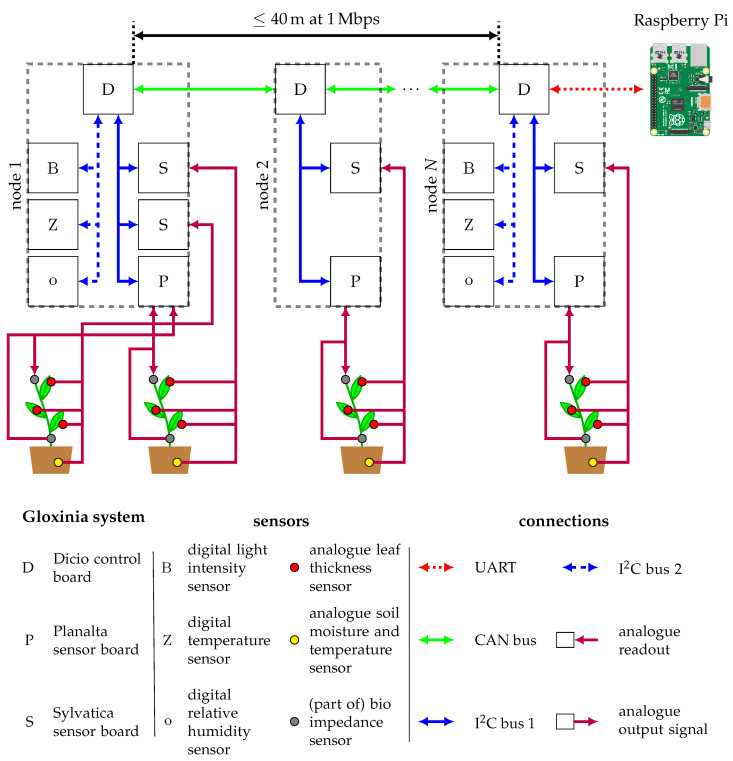
System architecture of a typical experiment where multiple plants are monitored using the same set of sensors. Analogue sensors are connected to Planalta and Sylvatica, depending on the type of sensor. For instance, a Linear Variable Displacement Transducer needs a variable input signal, so this type of sensor has to be connected to Planalta because this board can generate analogue signals. Sensors that only require an analogue readout are connected to Sylvatica. Environmental conditions are also measured with digital sensors at node 1 and node *N*. The data is transmitted and stored on a single board computer (Raspberry Pi) for further analysis. This figure is best viewed in colour. Raspberry Pi illustration by, Lucas Bosch, based on work by Jonathan Rutheiser [CC BY-SA] (https://creativecommons.org/licenses/by-sa/3.0).

**Figure 2 sensors-20-03055-f002:**
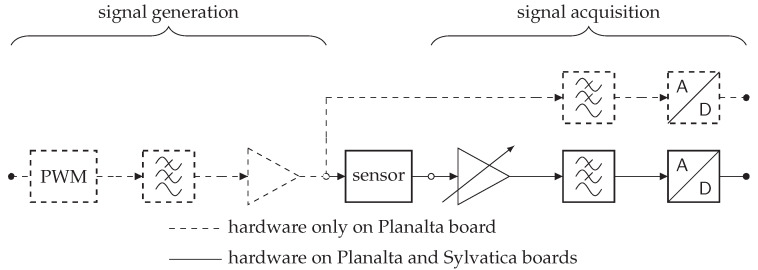
Block diagram of the analogue front-end for Sylvatica and Planalta boards. The Planalta-only driver generates a voltage wave that is generated through a pulse-width modulation (PWM) signal, which is low-pass filtered and then buffered before being fed into the sensor. The sensor output on both the Planalta and Sylvatica boards is (optionally) amplified using a Programmable Gain Amplifier (PGA) and low-pass filtered before being sampled.

**Figure 3 sensors-20-03055-f003:**
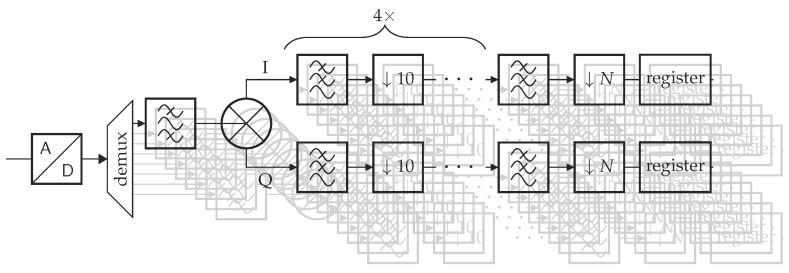
Schematic representation of the data processing and flow on the microcontroller of the Planalta board. The data coming from the analogue-to-digital converter ADC contains data from all channels. These samples are copied to new vectors by a “demultiplexer” (demux) to make the filtering faster. The incoming data is then band-pass filtered before being mixed to remove unwanted signals that the mixer can map to the same frequencies. Afterwards, the data passes through four filter and decimation stages before its final filtering and decimation step. The decimation factor *N* depends upon the incoming data rate to achieve an output frequency of 1 Hz.

**Figure 4 sensors-20-03055-f004:**
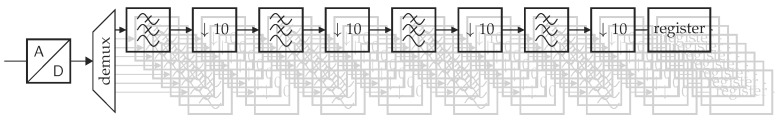
Schematic representation of the data processing and flow on the microcontroller of the Sylvatica board. The data coming from the analogue-to-digital converter ADC contains data from all channels. These samples are copied to new vectors by a “demultiplexer” (demux) to make the filtering faster. The filtering consists of a cascade of low-pass filters, followed by a decimation step of factor 10 to reduce the sample rate.

**Figure 5 sensors-20-03055-f005:**
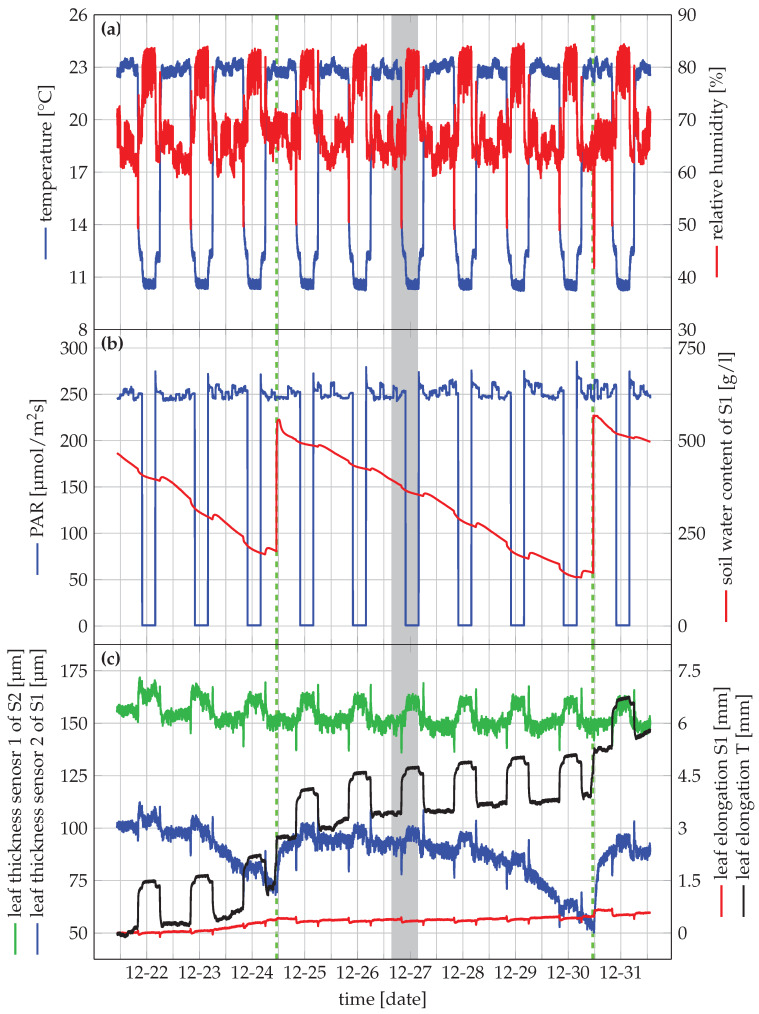
Visualisation of the captured data in a growth chamber experiment with a strawberry plant. The dashed green line indicates the watering time point. A detailed figure of the grey shaded line (16:30 on 26 December to 3:30 on 27 December) is depicted in Figure 7.

**Figure 6 sensors-20-03055-f006:**
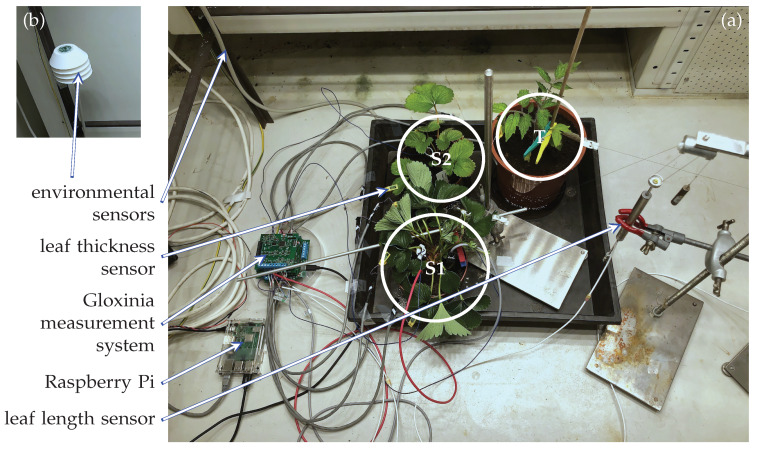
(**a**) Experimental set-up. The environmental sensors are not depicted in this figure, but their cable is. (**b**) Close-up of the radiation shield that houses the environmental sensors.

**Figure 7 sensors-20-03055-f007:**
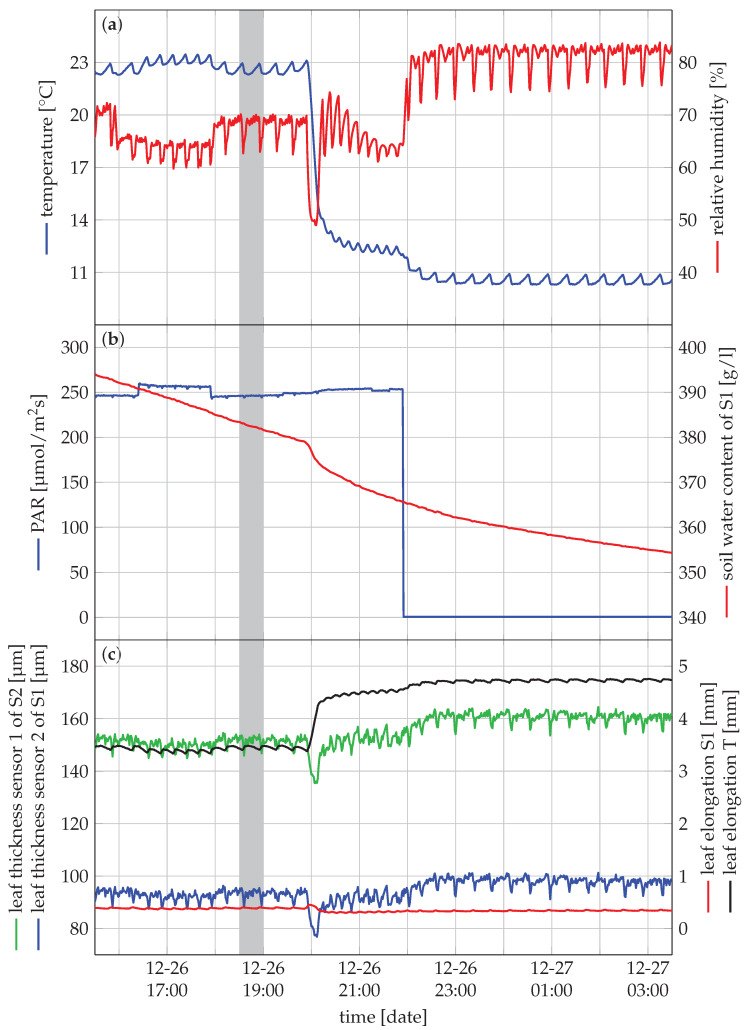
This figure is a detailed zoom of the grey shaded in Figure 5 to visualise the accuracy of the system. A further zoom of the grey shaded area in this figure (18:30 to 19:00 on 26 December) is depicted in Figure 8.

**Figure 8 sensors-20-03055-f008:**
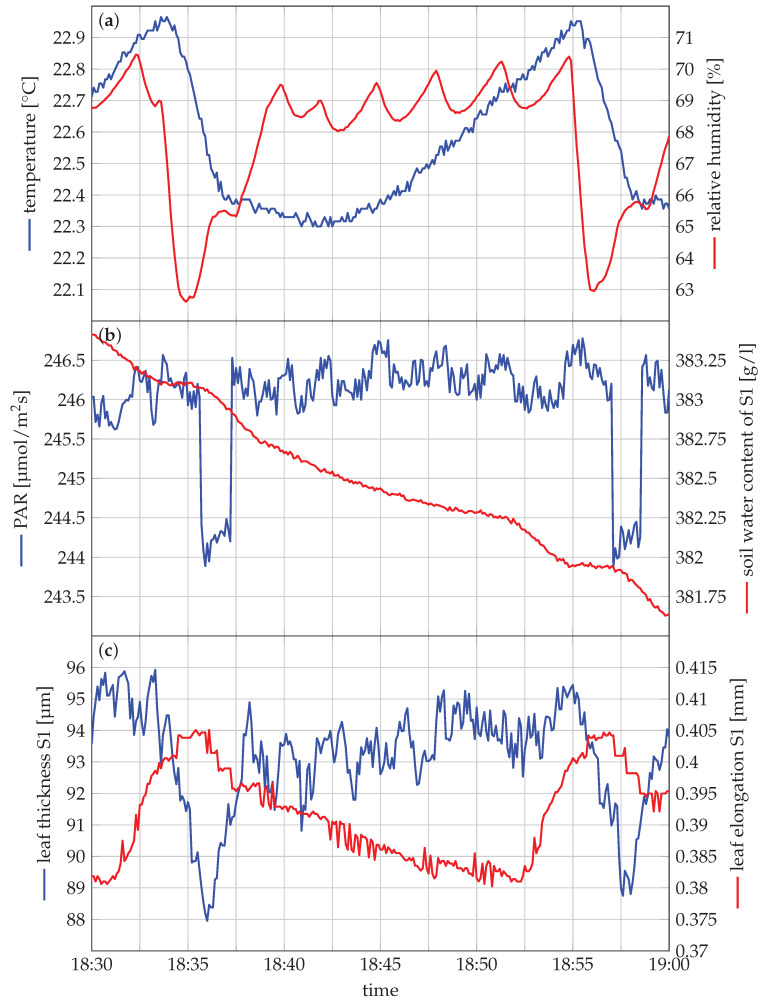
Detail figure of the grey shaded in Figure 7 to visualise of the accuracy of the system. Only the leaf thickness and leaf length of S1 are plotted since there is a significant offset between the leaf thickness of S1 and S2 and leaf length of S1 and T in Figure 7.

**Figure 9 sensors-20-03055-f009:**
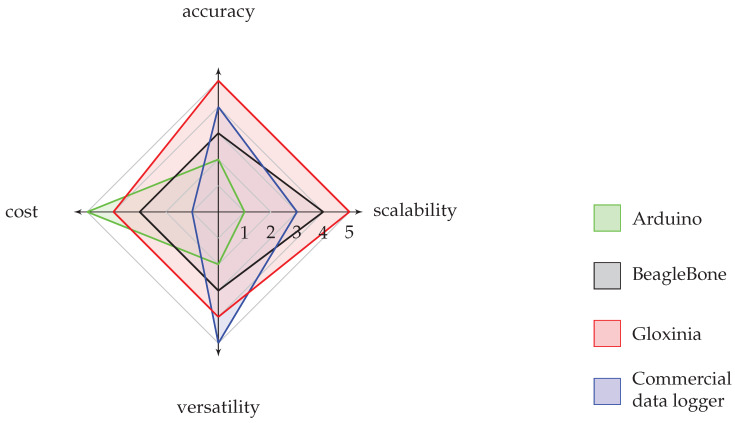
Spider chart comparing different data logging approaches. The higher the number, the better the performance for this specific metric. For instance, a score of 5 on the cost scale means this has the best cost performance, i.e., lowest cost.

**Table 1 sensors-20-03055-t001:** Overview of the different sampling and signal frequencies employed in the sensor boards Planalta and Sylvatica. $N_{c}$ is the maximum number of active channels, where “R” denotes that the output of the signal driver is also sampled (top analogue-to-digital converters (ADC) block in Figure 2). $f_{c}$ is the analogue cut-off frequency of the anti-aliasing low-pass filter in front of the ADC. $f_{s}$ is the sampling frequency of a specific ADC channel. The frequency depends upon the number of active channels. $f_{\mathrm{signal}}$ is the (modulated) signal frequency. The filter cascade reduces the input signal to a 1 kHz signal, as indicated by $f_{\mathrm{out}}$.

Board	$N_{c}$ [-]	$f_{c}$ [kHz]	$f_{s}$ [kHz]	$f_{\mathrm{signal}}$ [Hz]	$f_{\mathrm{out}}$ [Hz]
Dicio	-	-	-	-	-
Planalta	1	97.0	200	50 × 10${}^{3}$	1
	1 + R	39.3	80	20 × 10${}^{3}$	1
	2 + 2R	19.5	40	10 × 10${}^{3}$	1
	4 + 4R	10.3	20	5	1
Sylvatica	8	4.8	10	≤0.2	1

**Table 2 sensors-20-03055-t002:** Overview of the sensors connected to each plant. A leaf thickness sensor was connected at the start of the experiment to the tomato plant, but there was an issue with the connection. As a result, this data was not valid and not included in the analysis.

Plant	Sensor Type		
	Leaf Thickness	Leaf Length	Soil Moisture
strawberry 1 (S1)	2	1	1
strawberry 2 (S2)	2	0	0
tomato 1 (T)	0	1	0

**Table 3 sensors-20-03055-t003:** Overview of the different sensors used in the experimental validation of the system.

Sensor Description	Part Number and Manufacturer	Interface
leaf thickness	AH-303 (AgriHouse, USA)	analogue readout
leaf length (LVDT)	E100 (Chauvin Arnoux, France)	LIA readout
relative humidity and temperature	SHT35 (Sensirion AG, Switzerland)	digital I^2^C sensor
light intensity	APDS-9301-020 (Broadcom, USA)	digital I^2^C sensor

**Table 4 sensors-20-03055-t004:** Cost calculation assuming ten boards are produced of a particular type. The cost calculation is based on the Mouser inventory and component prices of 17 December 2019. Enclosure is optionally available from Hammond Manufacturing for 20.21 €. All prices are excluding VAT.

Board	Cost Category (€)									Total Cost (€)
	Capacitors	Resistors	Connectors	Microcontroller	User Interaction	Amplifiers	Voltage Reference	Other	ADC	
Dicio	3.16	0.74	10.43	6.68	5.09		4.64	1.9		32.64
Sylvatica	3.41	1.25	3.52	6.68	3.26	15.84	4.81	1.35	9.44	49.56
Planalta	6.47	2.03	2.96	6.68	3.26	16.44	4.81	5.25	9.44	57.34

**Table 5 sensors-20-03055-t005:** Detailed comparison and summary of the evaluation of the different sensor platforms. The cost values are based on the cost per sensor for the small first experiment. The following scores are given from lowest to highest for a particular metric: −, +/−, +, ++, and +++.

Criterion	Platform			
	Arduino	BeagleBone Black	Commercial	Gloxinia
scalability	−	++	+/−	+++
accuracy [bit]	10 (+/−)	12 (+)	12/13 (++)	16 (+++)
cost [per sensor, €]	3.76 (+++)	12.55 (+)	66.67–100 (−)	8.78 (++)
versatility	+/−	+	+++	++

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
