# Peer review of "Gloxinia—An Open-Source Sensing Platform to Monitor the Dynamic Responses of Plants"

_sensors, 2020, doi:10.3390/s20113055_

Round 1
Reviewer 1 Report
Dear Authors,
The manuscript is indeed fascinating, and I have to acknowledge that it is by default, more technical. I honestly had a hard time to go through every detail and understand all the information provided. From my personal experience, if I go through method or other technical papers to read, I do pay attention to certain parts of paper, like an introduction. I do feel, you have tried to highlight the importance of monitoring plant dynamics in very high temporal resolution. Unfortunately, you have provided insufficient evidence. It would be easier for the reader to follow if you could give few, aligned justifications of high temporal resolution monitoring. Stating that it is "crucial" several times and referring to other papers does not do a job for me (However, it is only my subjective opinion).
30- 31: The last sentence sounds a bit vague. "These" is not clear to me what parameters are you referring to?
84-95: it is not clear what sensors are you talking about. This paragraph needs some work; it instead brought confusion. Please start with explaining by details about the hardware you used then go to assembly/architecture
141 - 147: you have stated earlier that you provide universal, low cost and high accuracy way of monitoring plant dynamic with a high temporal resolution. If you want to talk about the trade-off in this paragraph, I do believe you should provide the cost of compromise in terms of quality. e.g. Giving up on the number of components, how does it affect the quality? Accuracy? Etc.
168 - 210: These reads for me as a technical guide of some tool. I can not insist on putting this part into appendices. However, I can not see how it fits in the main body of the manuscript.
Exciting results and beautiful graphs. I have indeed learned how can I monitor potential stress in plants. My thoughts went further, e.g. how it can be applied to big-scale monitoring? Hectares of crops, for example? I can hardly imagine that.
I would give a green light for publication of this manuscript. Although, I would ask authors to make it more informative and readable for a broader audience. I somehow think the manuscript is way too much heavy with technical details which could be moved as a supplement. Instead of the technical information, you could elaborate better on notes I have provided above.
Best Regards
Author Response
Dear reviewer
Thank you for reviewing our paper. Please see the attachment.
Kind regards
Olivier Pieters and co-authors

Reviewer 2 Report
The authors describe a sensing platform for monitoring rapid plant physiological and growth responses, conduct a simple validation exercise in a growth chamber, and evaluate their platform relative to others currently available for the same purpose. I am a plant biologist, not an engineer, so I leave commentary on the technical aspects of the system architecture to others. Data presented in Figures 5-8 show that the system as tested can record plausible plant responses to both longer term environmental perturbations (e.g., soil drying over days) and quite rapid and subtle changes (e.g., small changes in PAR and RH over minutes). There is a well-articulated need for a relatively inexpensive and open-source recording platform such as this and their methods and results are clearly presented and reasonably interpreted. This report therefore should be of broad interest to environmental plant biologists.
A more extensive trial of the platform would, however, make it much more compelling to plant biologists and likely increase the paper’s impact. Three plants grown for 10 days in a single growth chamber is about the minimum test one might imagine. Extending the trial to glasshouse or field grown plants would provide the quality of data probably required to convince potential users to adopt the system.
The weakest part of the paper is section 5.2, and the comparison to existing platforms. It is not clear how the values in Table 5 were arrived at (how does one platform get three +s and another get only one?). Nor is it clear how the mostly qualitative values in Table 5 are translated into the quantitative axis values shown in Figure 9. this does not strike me as being an entirely objective comparison.
Line 279: this reads as though temperature in the growth chamber was raised from 10o to 32o during the growth experiment. That is not shown in the figures and may refer only to the calibration exercise.
Author Response
Dear reviewer
Thanks you for reviewing our paper. Please see the attachment.
Kind regards
Olivier Pieters and co-authors
